# Interlaminar Shear Characteristics of Typical Polyurethane Mixture Pavement

**DOI:** 10.3390/polym14183827

**Published:** 2022-09-13

**Authors:** Guohua Gao, Min Sun, Chuanchang Xu, Guangzhen Qu, Yaohui Yang

**Affiliations:** 1Shandong Key Laboratory of Highway Technology and Safety Assessment, Jinan 250000, China; 2Shandong Hi-Speed Engineering Test Co., Ltd., Jinan 250098, China; 3School of Transportation Engineering, Shandong Jianzhu University, Jinan 250101, China; 4School of Civil Engineering and Architecture, University of Jinan, Jinan 250022, China; 5Shandong Hi-speed Group Co., Ltd., Innovation Research Institute, Jinan 250101, China

**Keywords:** interlaminar stress, shear characteristics, polyurethane mixture, shear stress, pavement structure

## Abstract

Polyurethane (PU) can be used as a road material binder, and its mechanical properties, durability, temperature stability, and other road performance metrics are good. However, the interlayer bonding between PU mixtures and asphalt mixtures is poor. The influence of the pavement structure, interlayer treatment scheme, load, and environmental factors on the interlayer shear characteristics of PU mixture composite pavement is analysed. Further, dynamic modulus, Hamburg rutting, accelerated loading, and inclined shear tests were conducted, and the typical PU mixture pavement shear stress was calculated. The interlaminar shear stress of double layer PU mixture pavement, polyurethane–asphalt composite pavement, and typical asphalt pavement were calculated. The results showed that the PU mixture has a low rutting deformation rate, stable mechanical properties, and strong resistance to the coupled action of temperature, water, and loading. The double-layer PU mixture structure has good water-temperature stability and fatigue resistance; however, freeze–thaw and accelerated loading cause great damage to the double-layer PU mixture structure. The residual shear strength ratio after freeze–thaw cycles and accelerated loading is only 50.3% and 35.6%, respectively, while the influence on the double-layer asphalt mixture structure is less. The theoretical calculation results of different pavement structures show that when the temperature increases from 10 °C to 50 °C, the interlaminar shear stress of polyurethane–asphalt composite pavement increases by about 20%. Additionally, the shear stress of pavement PU mixture pavement and typical asphalt pavement is mainly affected by load, and the temperature changes have an obvious effect on the interlayer shear stress of polyurethane–asphalt composite pavement. The calculated maximum shear stress of the three pavement structures with different working conditions is less than the interlaminar shear strength measured by the inclined shear test, indicating that the interlaminar treatment scheme of composite specimens can meet the shear resistance requirements of the three typical pavement structure types.

## 1. Introduction

Polyurethane (PU) is a generic term for a class of synthetic high molecular weight polymers with diverse forms and applicability. Their main chain contains repeated carbamate groups with good high/low-temperature resistance and mechanical properties [1]. Since the end of the 1970s, researchers studying pavement began to use one-component or two-component polyurethane to replace asphalt as the binder in pavement materials [2]. When PU is used to replace asphalt as a road material binder, it greatly reduces construction energy consumption and emissions, improving the mechanical properties, durability, temperature stability, and other road performances of the mixture [3,4,5]. However, the interlayer bonding between PU mixtures and asphalt mixtures is poor, and many early failures are caused by the poor adhesion between layers [6,7,8]. When the interlayer bonding of the pavement structure is poor, cracks may occur in the weak part of the mixture under the repeated action of vehicle loads, which could lead to displacement and wrapping issues, resulting in relative slippage between interlayers. The core drilling samples of composite pavement with such issues have shown that the interface between most PUMs and asphalt mixtures is smooth, as shown in Figure 1.

Thus, it is clear that interlayer adhesion is a technical issue of PU composite pavement that needs to be solved. However, current studies primarily focus on three aspects: functional pavement [9,10,11], pavement structural layers [12,13,14], and bridge deck pavement [15]. There are few papers that focus on the interlaminar shear characteristics of typical polyurethane (PU) mixture pavement. Ying et al. selected PU adhesive, polymer waterproof binder, and SBS (styrene–butadiene–styrene)-modified asphalt as three binders, and carried out research on the interlayer bonding adaptability of the PU mixture and asphalt mixture with the 45° inclined shear test and pull-out test [16]. Chen et al. studied the interface shear performance between porous polyurethane mixtures (PPMs) and asphalt mixtures [17]. Because the gradation of the PU mixture, bonding materials, environmental conditions, and many other factors affect the interlaminar shear properties, the interlaminar shear resistance characteristics remain uncertain during practical application. Therefore, the shear resistance characteristics of PU pavement structures should be considered to control premature PU pavement damage. 

Many studies have focused on the interlaminar shear properties of rigid-flexible composite pavement composed of an asphalt mixture and cement concrete [18,19]. Cao et al. divided the interlaminar shear and deformation curve into four stages: the elastic stage, failure stage, shear strength reduction phase, and residual phase [20]. Li et al. developed a device for interlaminar shear fatigue tests, determining the optimum asphalt dosage of interlaminar spraying structures in rigid-flexible composite pavement [21]. Wang et al. studied the influence of different cement concrete slab treatments, different waterproof adhesive materials, and different asphalt contents of the interface layer on the strength of the interfacial layer through indoor shear tests and pull-out tests on composite boards [22]. Mohammad et al. studied the influence of different factors on the peak value of the interlaminar shear strength according to field and indoor tests [23]. Luo et al. studied the influence of interface contamination and water saturation on the interlayer bonding properties [24]. The test methods and calculation theories of the influence of different factors on the interlaminar shear strength of rigid-flexible composite pavement are relatively mature.

Therefore, it is necessary to analyse the influence of the pavement structure, interlayer treatment scheme, load, and environmental factors on the interlayer shear characteristics of PU mixture composite pavement based on the mechanical properties and water-temperature stability of PU mixtures, combined with the results of inclined shear tests and theoretical calculations of the shear stress of typical pavement structures. This can provide a basis for effectively controlling and preventing the interlayer shear failure of PU mixture pavement.

## 2. Material Composition and Typical Pavement Structure

### 2.1. Raw Materials

A one-component PU binder, two-component PU were Wanhua Chemical Co., Ltd. (Yantai, China). The emulsified asphalt, SBS-modified asphalt, 70# base asphalt, and aggregates were Shandong Huarui Co., Ltd. (Zibo, China). The emulsified asphalt, SBS-modified asphalt, and 70# base asphalt meet the provisions of JTG F40-2004. The technical indexes of SBS-modified asphalt and 70# base asphalt were shown in Table 1. The one-component PU binder was a modified isocyanate containing a certain amount of terminal NCO group, which could react with water in the air. The components of the two-component PU were isocyanate and polyol, which needed to be premixed before use. 

### 2.2. Mixture Material Composition

Since the pavement structure upper layer mostly adopts the mixture with a nominal maximum particle size of 13.2 mm, and the middle layer mostly adopts the mixture with a maximum nominal particle size of 19.0 mm, the PU mixture was decided to have nominal maximum particle sizes of 13.2 mm (PUM-13) and 19.0 mm (PUM-20). The stone matrix asphalt with nominal maximum particle sizes of 13.2 (SMA-13) and the asphalt concrete with nominal maximum particle sizes of 19.0 (AC-20) were analyzed. The SMA-13 and AC-20 was mixed in accordance with JTG E20-2011. The PUM-10 and PUM-13 were prepared at room temperature, both the aggregates and binder do not need to be heated before mixing. After the aggregates and binder were mixed in the mixing pot for about one minute, the mineral powders were added and mixed for about three minutes. The mixture composition of the four mixtures are shown in Table 2. 

The mineral aggregate gradation design results of PUM-13, PUM-20, SMA-13, and AC-20 are shown in Figure 2.

### 2.3. Typical PU Mixture Pavement Structure

Compared to asphalt mixtures, PU mixtures have a significantly larger dynamic modulus and more significant elastic properties, which can appropriately reduce the thickness of pavement structure layers [25]. The PU mixture can be applied to every surface layer or to one surface layer; this leads to the design of the full-thickness PU mixture surface layer and composite surface structure composed of the PU mixture and asphalt mixture, respectively [26]. Relevant studies have shown that PU mixtures and cement stabilized macadam mixtures have good interlayer connection characteristics, which can effectively ensure the integrity of pavement structure [27]. Multi-layer cement stabilized macadam mixtures are often adopted in expressway pavement, at present. Therefore, multi-layer cement stabilized macadam mixtures are used as the pavement base for PU mixture pavement [28]. To fully understand the interlaminar shear characteristics and long-term service performance of PU mixture pavement, the typical asphalt pavement structure of expressways is analysed and compared. Previous studies have shown that the dynamic modulus and shear strength of PU mixtures are higher than that of SBS modified asphalt mixtures, so the PU mixtures is applied in the middle surface layer of polyurethane-asphalt composite pavement structure, which can take full use of the performance advantages of PU mixtures [25]. The three pavement structures studied in this paper are shown in Figure 3. Previous research shows that the adhesive strength between PU mixtures and PU mixtures is high, so no adhesive layer is designed between the two layers of PU mixtures [25]. The interlayer treatments are carried out between the asphalt mixture and PU mixture, and between the asphalt mixture and asphalt mixture; the specific treatment schemes are shown in Table 3. 

## 3. Test Scheme and Calculation Theory

### 3.1. Test Scheme

(1)Dynamic modulus test

The dynamic modulus test is carried out in accordance with T 0738-2011 of JTG E20-2011. The test specimen is a cylinder with a diameter of 150 mm and a height of 170 mm. The dynamic modulus and phase angle of PUM-13, PUM-20, SMA-13, and AC-20 are determined, and three specimens are tested for each mixture. The test temperature is adjusted between 10 °C and 50 °C, and the loading frequency is adjusted between 0.1 Hz and 25 Hz.

(2)Hamburg rutting test

The Hamburg rutting test is carried out in accordance with the provisions of AASHTO T 324-2014. The test is carried out in a 50° water bath environment. The width of the test wheel is 47 mm, the load is 705 N, and the loading rate is (52 ± 2) times/min. The maximum permissible loading iterations are 20000 cycles, and the maximum permissible rutting depth is 20 mm. The test is stopped when either of the above two conditions is met. The Hamburg rutting tests of PUM-13, PUM-20, SMA-13, and AC-20 are then carried out. The specimens are cylinders with a diameter of 150 mm and a height of 60 mm, which are formed by rotary compaction. There are two specimens in each group, and two specimens are tested for each mixture.

(3)Accelerated loading test

To evaluate the long-term performance and interlayer shear resistance characteristics of typical pavement structures under the coupled action of high temperatures, water, and loading, indoor accelerated loading tests are carried out on the three typical composite pavement structures shown in Table 3. The all-environment pavement accelerated loading system (ALT-S100) developed by Shandong Jiaotong University, Jinan, China is used for testing [29], as shown in Figure 4a. The wheel load is 1000 kg, the rolling speed is 4.5 km/h, the effective working length of rolling is 1 m, and the loading rate is 4000 times/h.

The composite rutting plate specimen is formed with two iterations of rolling. When the lower plate is removed, the bonding layer is treated, and then it is put into a double-layer rutting test mould to roll and form the superstructure. Three composite rut plates are formed for each structure and arranged in the effective working area [29]. A double-layer rut board is placed on each side of the effective working area, which mainly plays the role of fixing and cushioning, and no specific test is carried out on it; the layout of the test specimens is shown in Figure 4b. During the test, there is 50 ℃ water circulating in the test tank, and the liquid level is slightly higher than that at the top of the specimens, as shown in Figure 4c. The accelerated loading test does not commence until the specimens are insulated in the water tank for 8 h. After 20,000 test iterations, the rutting depth of the three specimens is tested at certain intervals of load times, and the test is terminated after more than 300,000 rolling iterations. 

(4)Interlaminar inclined shear test

Previous studies have shown that the direct shear method cannot simulate practical road conditions. Herein, the friction coefficient (δ) between the wheel and the road surface is taken as 0.5, considering the horizontal load generated by the emergency braking of driving vehicles. Further, considering special circumstances, such as the emergency braking of vehicles in downhill sections, the 45° inclined shear test is more practical [27,28]. The 45° inclined shear test device developed by the project team was introduced in previous study [25,30]. The inclined shear strength is calculated using Equation (1), where τ represents the shear interlaminar strength of the interface, *F* represents the maximum vertical load, and *S* represents the contact area between two layers.
(1)$$\tau =\frac{F\cdot sin45^{{}^{\circ}}}{S}$$

The inclined shear specimens are cut from the composite rut plate. To evaluate the interlaminar shear characteristics of typical composite pavement structures under different working conditions, the specimens at room temperature (15 °C), with freeze–thaw cycles and after accelerated loading tests, are tested. The freeze–thaw conditions are implemented in accordance with the provisions of T 0729-2000. The specimens with accelerated loading tests are cut from the wheel trace belt position of the double-layer rutting plates after 300,000 loading iterations. The size of the inclined shear specimen is 50 mm × 50 mm × 100 mm, and five parallel specimens are tested under each test condition, whose average value is taken as the test result [30,31].

### 3.2. Theoretical Calculation

The numerical models of the three aforementioned pavement structures are established by using the ABAQUS finite element software, whose model size is 10 m (length) × 6 m (width) × 6 m (height). The elastic layered system is used to calculate the interlayer shear stress. It is assumed that each structural layer is homogeneous, completely elastic, and isotropic. The structural layers of the model are connected by bonding. The three-dimensional eight-node linear hexahedron element of C3D8R is used for mesh construction. The parameters of asphalt and PU mixtures are tested using the dynamic modulus test, and the material parameters of each structural layer are shown in Table 4. During the analysis, the boundary conditions of the model are as follows: the subgrade bottom is completely fixed, the road laterally constrains the displacement in the Z direction, longitudinally constrains the displacement in the X direction, and the Y axis is the direction of road thickness. The grid is densified in the load action area. The road structure calculation model is shown in Figure 5.

The interlaminar shear stress of the three typical pavement structures under the four working conditions described in Table 5 are calculated. In the theoretical calculation process, the contact shape between each wheel and pavement is equivalent to a rectangle, and the equivalent length and width of the rectangular action area are 22.65 cm and 15.6 cm, respectively. 

## 4. Results and Discussion

### 4.1. Dynamic Modulus Test Results

The dynamic modulus test results of the four mixtures at different test temperatures are shown in Figure 6.

The dynamic modulus curves at different temperatures reflect the following phenomena:

At the same temperature, the dynamic modulus of the four mixtures increases with increases in the loading frequency, and the dynamic modulus of SMA-13 and AC-20 increase obviously with increases in the frequency; the influence of the loading frequency on PUM-13 and PUM-20 is relatively small. When the loading frequency is greater than 5 Hz, the dynamic modulus curves of PUM-13 and PUM-20 are close to the horizontal line, indicating that the dynamic mechanical properties of polyurethane mixtures are stable.

Under the same loading frequency, the higher the temperature, the smaller the dynamic modulus. That is, increases in the ambient temperature will reduce the ability of the mixture to bear the load, and the influence of temperature on the dynamic modulus of asphalt mixtures is much greater than that of PU mixtures [32,33]. Taking the dynamic modulus under a 10 Hz loading frequency at 20 °C and 50 °C as an example, the 50 °C dynamic modulus of SMA-13 and AC-20 decreased by 94.2% and 93.2%, respectively, in comparison to the dynamic modulus at 20 °C, and those for PUM-13 and PUM-20 decreased by 28.1% and 29.8%, respectively. This is primarily because the isocyanate group of the PU binder reacts with the moisture in the air and the active hydrogen on the aggregate surfaces, forming a cross-linked network structure, which ensures the strength and temperature stability of PU mixtures [34,35].

Since vehicle driving speeds are generally between 60–120 km/h, and the 10 Hz loading frequency is the closest to actual road stress conditions, the phase angles of the four mixtures at different temperatures under 10 Hz, and the phase angles at different loading frequencies at 20 °C, are analysed and shown in Figure 7. 

It indicates that, under different ambient temperatures and loading frequencies, the phase angles of PUM-13 and PUM-20 are about 5°, while the phase angles of SMA-13 and AC-20 mixtures vary between 20°~35°, exhibiting a downward trend with increases in the loading frequency. This indicates that, within the pavement materials service temperature of 10–50 °C, the loading frequency and ambient temperature have little influence on the phase angle of PU mixtures, the viscoelastic properties of PU mixtures are stable, and the elastic characteristics are significant. Therefore, it is appropriate to use elastic layered system theory to simulate and calculate the load response of PU composite pavement [36]. 

### 4.2. Hamburg Rutting Test Results

The rutting depth of Hamburg rutting tests are shown in Figure 8. 

Figure 8a shows that the four mixtures do not reach the spalling turning point when the loading iterations reach 20000 cycles, that is, there is no spalling section. The rutting depths of SMA-13 and AC-20 are 4 mm and 7 mm, respectively, which meets the Hamburg rutting test index requirements of AASHTO T324 for PG 64 and PG 82 grade asphalt binders [36]. The rutting depth of the four mixtures after 10 thousand cycles and 20 thousand cycles rutting is shown in Figure 8b. When PUM-13 and PUM-20 are loaded 10 thousand times, the rutting depths are 0.69 mm and 0.74 mm, respectively, and the rutting depths at 20 thousand times are 0.8 mm and 0.7 mm, respectively, which is far less than those of SMA-13 and AC-20. Further, the creep slope of PUM-13 and PUM-20 mixtures are about 8%–12% of those of SMA-13 and AC-20 mixtures. The combination of water, temperature, and loading has a different function on the deformation characteristics of PU mixtures and asphalt mixtures [37]. The PU mixtures have low rutting deformation rates; strong resistance to the coupled action of temperature, water, and loading; and relatively stable structures. This is mainly due to the formation of urea bonds, carbamate bonds, and other macromolecular bonding network structures between the PU binder and aggregate, which are less affected by temperatures and loading.

### 4.3. Accelerated Loading Test Results

#### Rutting Depths of the Structures

The changes in the rutting depth of the three composite structures with long-term high-temperature water environments and loading are shown in Figure 9.

According to Figure 9, in a high-temperature water area under loading, the rutting depth of PUM-13+PUM-20 (Structure-I) was the smallest, which increased very slowly. Until the accelerated loading test was completed after 300 thousand cycles loads, the maximum rutting depth of Structure I; was only about 0.2 cm, and there were no cracks in the specimens (Figure 10a). This indicates that the pavement structure with a double-layer PU mixture had good water-temperature stability and fatigue resistance. This is because the curing reaction of –NCO with the moisture in the air, forming a cross-linked structure—including urea bonds, amino formate bonds, and urea formate bonds. The reaction effectively ensure the water-temperature stability and durability of composite pavement structures with double-layer PU mixtures. 

However, the rutting depth of the typical asphalt pavement structure (Structure-III) changed the fastest. After 300,000 loading cycles, the rutting depth was found to be greater than 1.2 cm, and there were many micro-cracks in the specimen (Figure 10c), indicating that the water-temperature stability and fatigue resistance of double-layer asphalt mixture structures was worse than those of other structures [38]. This was mainly caused by the strong temperature sensitivity of asphalt materials.

For the composite pavement structure (Structure-II), the development speed of the rutting depth was between those of Structure-I and Structure-III. At the end of the loading test, the maximum rutting depth of the specimen was 0.8 cm, and only a few micro-cracks appeared in the specimen (Figure 10b). This indicated that the composite structure composed of an asphalt mixture and a PU mixture could reduce the rutting depth—to a certain extent—and improve the water stability and durability of the composite pavement structure, primarily because the PU mixture in the lower layer had a high dynamic modulus and strength. 

### 4.4. Inclined Shear Test Results under Different Working Conditions

The interlaminar shear strength test results of the three composite specimens under different working conditions are shown in Figure 11.

#### According to the data in Figure 11

At 15 °C (room temperature), the interlaminar shear strength of structure-I is 1.71 and 1.79 times that of structure-II and structure-III, respectively, that is, the double-layer PU mixture structure has good shear properties at room temperature. However, after the accelerated loading test and freeze–thaw test, the shear strength of structure-I decreases significantly, and the residual shear strength ratios after freeze–thaw and accelerated loading are only 50.3% and 35.6%, respectively. This shows that freeze–thaw and accelerated loading greatly reduce the shear strength of the structure of double-layer PU mixtures. This may be because, after freeze–thaw, high temperature, and loading, the weak points (C–O bonds, unsaturated C=C double bonds, etc.) in the interlayer molecular chain of the PU mixture may experience chemical changes, such as bond breaking, which may affect the interlayer shear properties.

The interlaminar shear strength of structure-II under three working conditions is lower than that of structure-I and structure-III, and the freeze–thaw and accelerated loading have a great impact on the shear strength of structure-II. The residual shear strength ratio after freeze–thaw and accelerated loading is only 56.3% and 24.6%, respectively. This is because the different material characteristics of asphalt mixtures and PU mixtures lead to different stress transmission and strain development of the two materials, which leads to interlaminar issues. Therefore, when this structure is applied to engineering, it is necessary to carefully design the interlayer treatment scheme [25].

The interlaminar shear strength of structure-III under the three working conditions is similar, and the residual shear strength ratio after freeze–thaw and accelerated loading are 84% and 97.1%, respectively, indicating that the double-layer asphalt mixture pavement structure can still maintain good overall bonding after freeze–thaw, high-temperature water-accelerated loading. The structure of double-layer asphalt mixture has good interlayer stability.

### 4.5. Calculation Results for the Interlaminar Shear of Typical Pavement Structures

#### 4.5.1. Distribution of Shear Stress Values in the Depth Direction 

The shear stress values in the depth direction of three typical pavement structures under the four working conditions are shown in Figure 12.

The shear stress of the three pavement structures increases first and then decreases along the pavement depth direction. Under working condition IV, i.e., 0.9 MPa loading and 50 °C temperature, the interlayer shear stress of the three pavement structures reaches its maximum value, where the maximum shear stress of structure-I and structure-III appears at the bottom of the upper layer, and the maximum shear stress of structure-II appears at the bottom of the middle layer. The reason may be that the material characteristics of the upper and middle layers of structure-I and structure-III are the same, but the material characteristics of the upper and middle layers of structure-II are different, and the modulus of the middle layer material is higher than that of the upper layer, resulting in the change of the position of maximum shear stress.

Comparing the shear stress calculation results under the two load levels of 10 °C and 50 °C, it can be seen that the shear stress calculation result under 0.9 MPa is significantly greater than that under 0.7 MPa, and the influence of loads on the shear stress of different structures is similar, indicating that overload will greatly increase the shear stress of the pavement structure.

Comparing the calculation results of shear stress at 10 °C and 50 °C under loads of 0.7 MPa and 0.9 MPa, for structure-I and structure-III, the shear stress with different temperatures is similar under the same load level; however, the law for structure-II is different. The shear stress of 50 °C is greater than that at 10 °C. This is mainly because the modulus of asphalt mixture decreases rapidly and the viscosity characteristic is more obvious under the action of high temperature, while the modulus of polyurethane mixture changes less and still maintains the elastic characteristic.

#### 4.5.2. Layer Bottom Shear Stress of the Typical Pavement Structure

The calculation results for interlayer shear stress at the bottom of the upper layer of three typical pavement structures under the four working conditions are shown in Figure 13.

Along the transverse direction of the road, at 0.3–1.2 m and 4.8–5.7 m, the shear stress is almost zero, that is, the tire has little impact on the road surface within this range. The shear stress begins to increase gradually at 1.2–1.5 m and 4.5–4.8 m, and reaches a maximum at the tire grounding edge. Then, the shear stress decreases and becomes zero at the symmetrical positions of both tires.

When the driving load is the same, the shear stress transverse distribution curves of structure-I and structure-III under different temperatures are similar; however, the law for structure-II is different. For structure-II, when the temperature increases from 10 °C to 50 °C, the interlaminar shear stress increases by about 20%. This shows that the shear stress of structures-I and structures-III are mainly affected by loading, and the temperature change has an obvious effect on structure II’s interlayer shear stress.

The upper layer bottom maximum shear stress results of the three pavement structures are summarized in Table 6.

The maximum shear stress produced by structure-I in the two working conditions at 10 °C is greater than those of the other two structures, and the maximum shear stress of structure-II for the two working conditions at 50 °C is greater than those of the other two structures. Compare the maximum shear stress of produced by the three pavement structures with the shear strength of different structures shown in Figure 11. The maximum shear stress is less than the interlaminar shear strength measured, it indicates that the shear strength of the composite specimen under different environment and load conditions can meet the requirements of pavement structures, that is, the proposed interlaminar treatment scheme is reasonable.

## 5. Conclusions

The PU mixture has a low rutting deformation rate and strong resistance to the coupled action of temperature, water, and loading, and the elastic characteristics of the PU mixture are more obvious than the SBS modified asphalt mixture.

The double-layer PU mixture structure has better water-temperature stability and fatigue resistance than double-layer asphalt mixture structures. The freeze–thaw action and accelerated loading cause significant damage to the double-layer PU mixture structure. However, the shear strength of the double-layer PU mixture structure with different working conditions is still higher than the shear stress of the PU pavement structure. Overloading greatly increases pavement structure shear stress, and temperature change has a great impact on the shear stress of polyurethane-asphalt composite structures. The calculated maximum shear stress of the three pavement structures under different working conditions is less than the interlaminar shear strength measured by the inclined shear test, thus the proposed three typical pavement structures are appropriate.

The authors conducted experimental research and theoretical calculation on the shear resistance of the three typical structures. However, in order to fully grasp the performance of PU mixture pavement, it is still necessary to carry out more research on various load responses of different types of PU mixture pavement in the future.

## Figures and Tables

**Figure 1 polymers-14-03827-f001:**
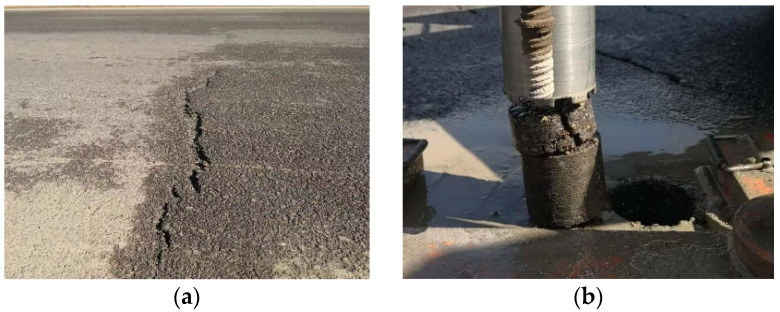
Interlayer slippage of PU mixture composite pavement. (**a**) Interlaminar slippage. (**b**) Interlaminar shedding.

**Figure 2 polymers-14-03827-f002:**
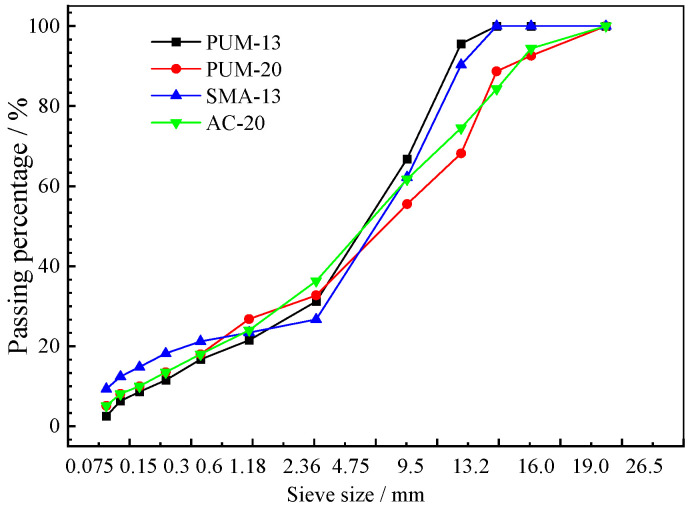
Mineral aggregate gradation design results.

**Figure 3 polymers-14-03827-f003:**
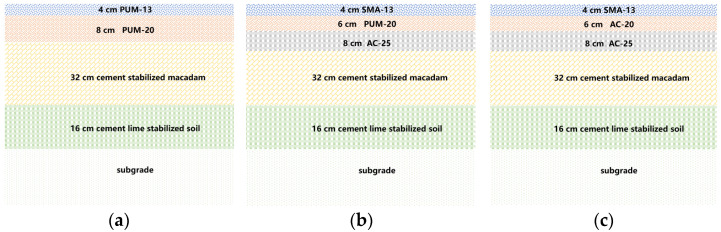
Typical PU pavement and asphalt pavement structure. (**a**) Double layer PU mixture pavement. (**b**) Polyurethane–asphalt composite pavement structure. (**c**) Typical asphalt pavement.

**Figure 4 polymers-14-03827-f004:**
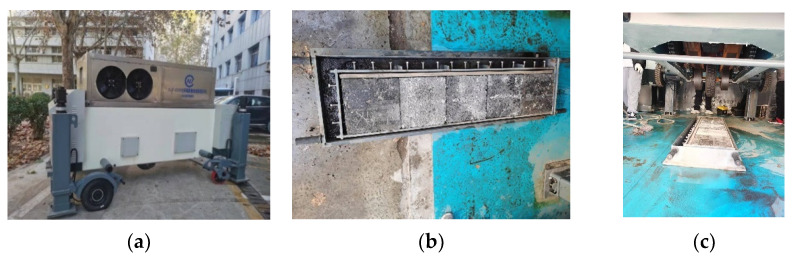
Experimental equipment and test conditions. (**a**) Test device. (**b**) Layout of specimens. (**c**) Loading test process.

**Figure 5 polymers-14-03827-f005:**
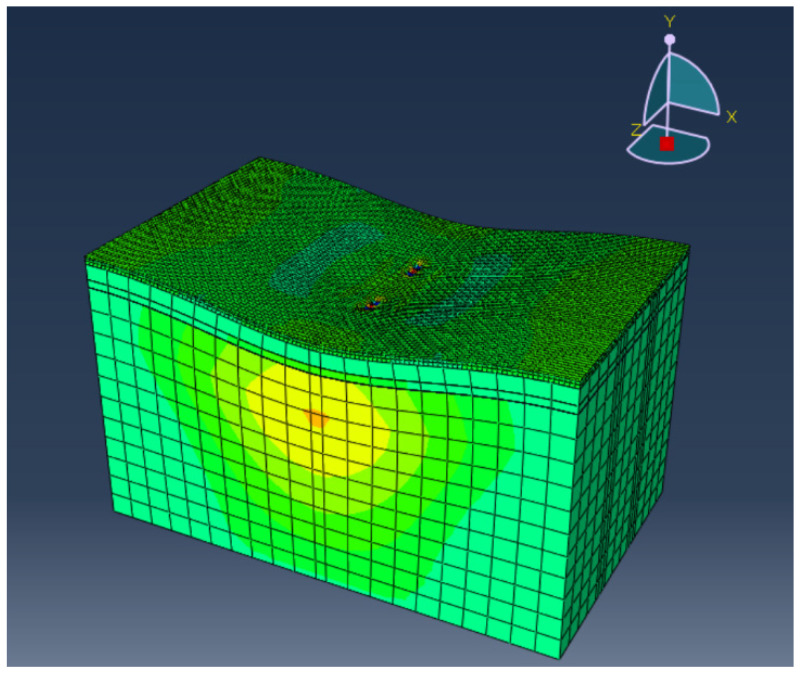
Calculation model for the road structure.

**Figure 6 polymers-14-03827-f006:**
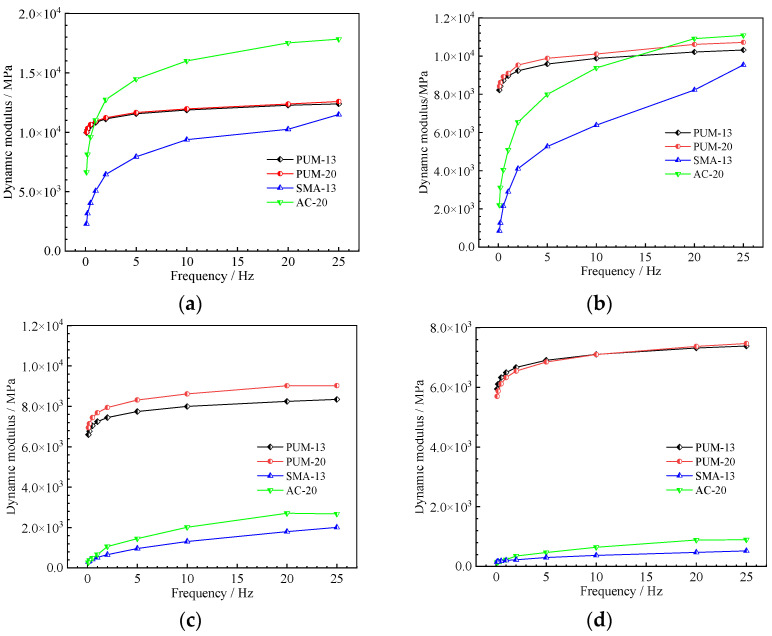
Dynamic modulus of the four mixtures. (**a**) 10 °C. (**b**) 20 °C. (**c**) 35 °C. (**d**) 50 °C.

**Figure 7 polymers-14-03827-f007:**
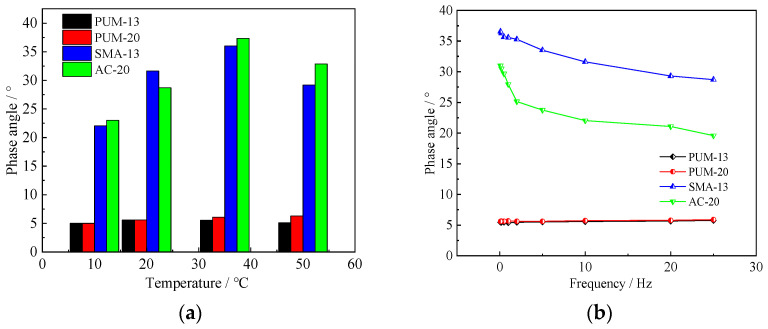
Phase angles of the four mixtures. (**a**) Dynamic modulus at 10 Hz frequency. (**b**) Dynamic modulus at 20 °C.

**Figure 8 polymers-14-03827-f008:**
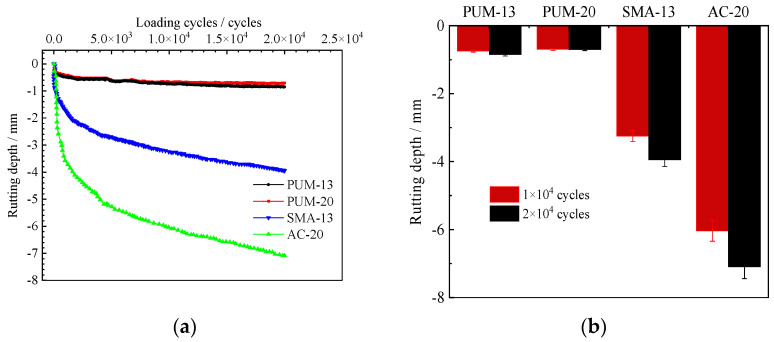
Hamburg rutting test results of the four mixtures. (**a**) Hamburg rutting curves. (**b**) Rutting depth after the Hamburg rutting test.

**Figure 9 polymers-14-03827-f009:**
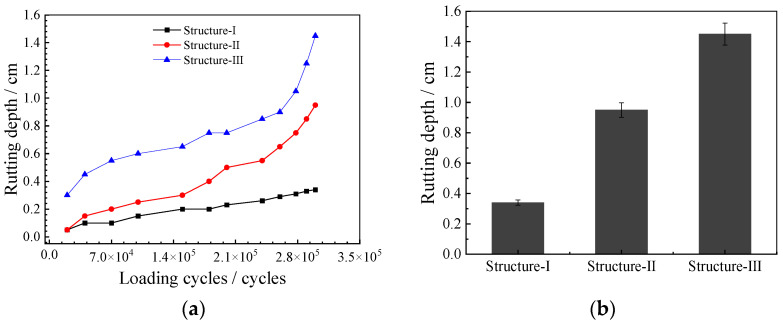
Rutting depth of the three composite specimens. (**a**) Rutting depth development curves. (**b**) Rutting depth after accelerated loading tests.

**Figure 10 polymers-14-03827-f010:**
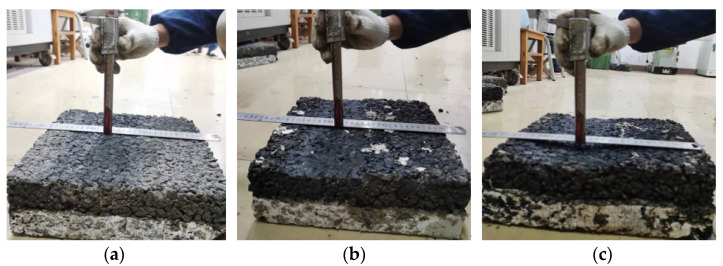
Images of the different pavement structures. (**a**) Structure-I. (**b**) Structure-II. (**c**) Structure-III.

**Figure 11 polymers-14-03827-f011:**
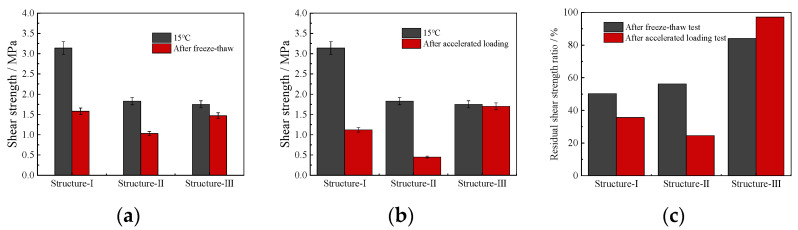
Shear strength under different working conditions. (**a**) After freeze–thaw. (**b**) After accelerated loading. (**c**) Residual shear strength ratio.

**Figure 12 polymers-14-03827-f012:**
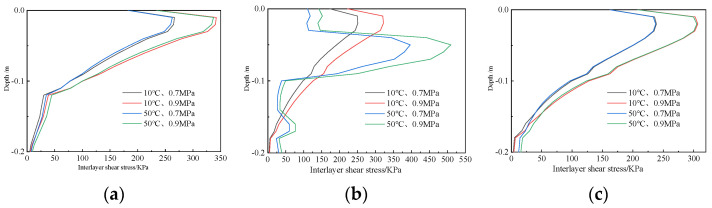
Shear stress distributed along the depth direction. (**a**) Structure-I. (**b**) Structure-II. (**c**) Structure-III.

**Figure 13 polymers-14-03827-f013:**
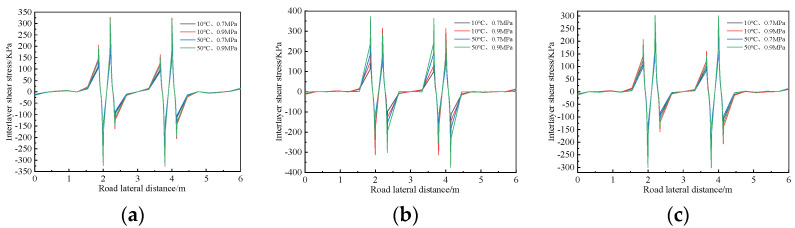
Shear stress distribution at the bottom of the upper layer. (**a**) Structure-I. (**b**) Structure-II. (**c**) Structure-III.

**Table 1 polymers-14-03827-t001:** The indexes of SBS-modified asphalt and 70# base asphalt.

Technical Indicators	Test Method	Unit	SBS Modified Asphalt	70# Base Asphalt
Penetration	T 0604	0.1 mm	53	71
Penetration Index	T 0604	/	−0.5	0.9
Ductility	T 0605	cm	48 (5 °C)	83 (15 °C)
Softening Point	T 0606	°C	86.5	48.5
Dynamic Viscosity of 135 °C	T 0625	Pa.s	2.16	0.25

**Table 2 polymers-14-03827-t002:** The mixture composition of the four mixtures.

Binder	Binder Content/%	Nominal Maximum Particle Sizes/mm	Representative Symbols
polyurethane	5.1	13.2	PUM-13
polyurethane	5.0	19.0	PUM-20
SBS modified asphalt	5.8	13.2	SMA-13
70# base asphalt	4.8	19.0	AC-20

**Table 3 polymers-14-03827-t003:** Composite specimens of the three pavement structures.

Serial Number	Structure-I	Structure-Ⅱ	Structure-Ⅲ
Upper layer	PUM-13	SMA-13	SMA-13
Adhesive layer material	/	two-component PU	emulsified asphalt
Content/(L/m^2^)	/	0.4	0.5
Lower layer	PUM-20	PUM-20	AC-20

**Table 4 polymers-14-03827-t004:** Material parameters of each structural layer.

Material	Dynamic Modulus (MPa)		Poisson’s Ratio
	10 °C	50 °C	
PUM-13	11,869	7101	0.25
PUM-20	11,967	7095	0.25
SMA-13	9372	370.3	0.25
AC-20	16,000	642.1	0.30
AC-25	17,000	682.2	0.30
Cement stabilized macadam	16,000	16,000	0.25
Cement lime stabilized soil	12,000	12,000	0.25
Subgrade soil	70	70	0.40

**Table 5 polymers-14-03827-t005:** Working condition specifications.

Working Condition	Temperature/°C	Axle Load/MPa
Ⅰ	10	0.7
Ⅱ	10	0.9
Ⅲ	50	0.7
Ⅳ	50	0.9

**Table 6 polymers-14-03827-t006:** Upper layer bottom maximum shear stress results.

Structure	Maximum Shear Stress/MPa			
	Working Condition I (10 °C, 0.7 MPa)	Working Condition II (10 °C, 0.9 MPa)	Working Condition III (50 °C, 0.7 MPa)	Working Condition IV (50 °C, 0.9 MPa)
I	0.2544	0.3271	0.2484	0.3194
II	0.2442	0.3140	0.2916	0.3749
III	0.2347	0.3018	0.2337	0.3004

## Data Availability

The data used to support the findings of this study are available from the corresponding author upon request.

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
