# Peer review of "Interlaminar Shear Characteristics of Typical Polyurethane Mixture Pavement"

_polymers, 2022, doi:10.3390/polym14183827_

Round 1
Reviewer 1 Report
The paper entitled “Interlaminar shear characteristics of typical polyurethane mixture pavement” is a valid research work with appropriate experimental evidence. The topic of the manuscript is of interest. The introduction section represents the state-of-the-art in the specific field of research and is appropriately supported by the relevant references. The results are described and treated correctly. However, the technical quality of the manuscript is average. The authors are requested to make changes according to the comments below before the paper can be accepted for publication.
- In the abstract, line 29, the author mentioned structure I and structure II; I will suggest the author, instead of saying structure I and structure II, say the difference between two structures I and II and their consequent result.
- I will suggest the author to mention how they determined the size of the particle (Table 1)
- Please do modify all the figures. For example, mark figures a & b, and in the figure caption, mention all the text.
- The conclusion is not well written. I will suggest the author to rewrite the conclusion.
Author Response
The paper entitled “Interlaminar shear characteristics of typical polyurethane mixture pavement” is a valid research work with appropriate experimental evidence. The topic of the manuscript is of interest. The introduction section represents the state-of-the-art in the specific field of research and is appropriately supported by the relevant references. The results are described and treated correctly. However, the technical quality of the manuscript is average. The authors are requested to make changes according to the comments below before the paper can be accepted for publication.
- In the abstract, line 29, the author mentioned structure I and structure II; I will suggest the author, instead of saying structure I and structure II, say the difference between two structures I and II and their consequent result.
Response:Thank you for your comment. The authors have revised the abstract.
The interlaminar shear stress of double layer PU mixture pavement, polyurethane–asphalt composite pavement and typical asphalt pavement were calculated. (See Line 23-25.)
The theoretical calculation results of different pavement structures show that, when the temperature increases from 10 ℃ to 50 ℃, the interlaminar shear stress increases of polyurethane–asphalt composite pavement by about 20%. And the shear stress of pavement PU mixture pavement and typical asphalt pavement is mainly affected by load, and the temperature changes have an obvious effect on the interlayer shear stress of polyurethane–asphalt composite pavement. (See Line 31-35)
- I will suggest the author to mention how they determined the size of the particle (Table 1)
Response:Thank you for your comment. The authors have revised the introduction.
Since the pavement structure upper layer mostly adopts the mixture with nominal maximum particle sizes of 13.2 mm, and the middle layer mostly adopts the mixture with the maximum nominal particle size of 19.0 mm, the PU mixture was detected to have nominal maximum particle sizes of 13.2 mm (PUM-13) and 19.0 mm (PUM-20), and the stone matrix asphalt, with nominal maximum particle sizes of 13.2 (SMA-13), and asphalt concrete with nominal maximum particle sizes, of 19.0 (AC-20), were analyzed. (See Line 111-113)
- Please do modify all the figures. For example, mark figures a & b, and in the figure caption, mention all the text.
Response:Thank you for your comment. The authors have revised the figures. (See Line 60, Line 196, Line 245, Line 271, Line 284, Line 308, Line 321, Line 341, Line 376, Line 406)
- The conclusion is not well written. I will suggest the author to rewrite the conclusion.
Response:Thank you for your comment. The authors have rewritten the conclusion.
(See Line 432-448)
Reviewer 2 Report
In this paper the authors have studied the influence of the pavement structure, interlayer treatment scheme, load, and environmental factors on the interlayer shear characteristics of PU mixture composite pavement structures. The paper can be accepted after minor revisions:
1. The authors have studied three pavement structure types in the paper. Is there a reason why a structure with upper layer as PU-13 and second layer as AC-20 was not included?
2. Figures in results and discussion section must be improved to make them the same size. Also figure descriptions must be improved
3. Conclusions could be framed much better
Author Response
In this paper the authors have studied the influence of the pavement structure, interlayer treatment scheme, load, and environmental factors on the interlayer shear characteristics of PU mixture composite pavement structures. The paper can be accepted after minor revisions:
- The authors have studied three pavement structure types in the paper. Is there a reason why a structure with upper layer as PU-13 and second layer as AC-20 was not included?
Response:Thank you for your comment. The authors have explained the reason in detail.
Previous studies have shown that the dynamic modulus and shear strength of PU mixtures are higher than that of SBS modified asphalt mixtures, so the PU mixtures is applied in the middle surface layer of polyurethane-asphalt composite pavement structure, which can take full use of the performance advantages of PU mixtures. (See Line 117-121)
- Figures in results and discussion section must be improved to make them the same size. Also figure descriptions must be improved.
Response:Thank you for your comment. The authors have revised the figures and figure descriptions in results and discussion section.
(See Line 245-247, Line 271-272, Line 285-286, Line 291-293, Line 308-310, Line 321-323, Line 368-369)
- Conclusions could be framed much better
Response:Thank you for your comment. The authors have rewritten the conclusion.
(See Line 432-448)
Reviewer 3 Report
Attractive manuscript related to the assessment of PUM (Polyurethane asphalt mixtures) with NMAS (nominal maximum aggregate size) of 13 and 20 mm.
In general, the paper is well presented, but some details must be clarified or corrected/completed:
1. Line 98: please define “70# base asphalt”;
2. Lines 100/103: considering these lab mixing conditions, how can these operations be carried out in an asphalt plant?
3. Section “2.1. Raw materials”: you can also include the main properties for “70# base asphalt” and “SBS-modified asphalt”;
4. Line 106: “stone matrix asphalt” (USA) or “stone mastic asphalt” (Europe);
5. Figure 3 (a)/Table 2: why didn't you include a tack-coat between PUM-13 and PUM-20 (Structure-I)?
6. Line 164: can you include some references about the “accelerated loading system”?
7. Section “4. Results and Discussion”: you should include some further discussion in light of results obtained by other researchers;
8. Figure 8 (b): please give note that these values were obtained after 2.0E+4 cycles;
9. Line 272: “Figure 11” or “Figure 8”?
10. Figure 9: the unit of “Rutting depth” is “cm” instead of “mm”;
11. Section “5. Conclusions”: you can still include or develop further conclusions. At the end of this section (or in section 4), you can also add final comments, like some information on costs and further works;
12. A final comment: some of the information/images presented in this manuscript have already been included in another (“Interlaminar Shear Characteristics, Energy Consumption, and Carbon Emissions of Polyurethane Mixtures”, by “Yufeng Bi, Min Sun, Shuo Jing, Derui Hou, Wei Zhuang, Sai Chen, Xuwang Jiao and Quanman Zhao” in “https://doi.org/10.3390/coatings12030400”), so I suggest that you reformulate, as far as possible, some content.
Author Response
Attractive manuscript related to the assessment of PUM (Polyurethane asphalt mixtures) with NMAS (nominal maximum aggregate size) of 13 and 20 mm. In general, the paper is well presented, but some details must be clarified or corrected/completed:
- Line 98: please define “70# base asphalt”;
Response:Thank you for your comment. The authors have defined 70# base asphalt and SBS modified asphalt. The technical indexes of SBS-modified asphalt and 70# base asphalt were shown in Table 1. (See Line 109)
- Lines 100/103: considering these lab mixing conditions, how can these operations be carried out in an asphalt plant?
Response:Thank you for your comment. The authors have revised and given clear description.
The SMA-13 and AC-20 was mixed in accordance with JTG E20-2011. The PUM-10 and PUM-13 were prepared at room temperature, both the aggregates and binder do not need to be heated before mixing. After the aggregates and binder were mixed in the mixing pot for about one minutes, then the mineral powders were added and mixed for about three minutes. (See Line 117-121)
- Section “2.1. Raw materials”: you can also include the main properties for “70# base asphalt” and “SBS-modified asphalt”;
Response:Thank you for your comment. The authors have given the main properties of 70# base asphalt and SBS modified asphalt. The technical indexes of SBS-modified asphalt and 70# base asphalt were shown in Table 1. (See Line 109)
- Line 106: “stone matrix asphalt” (USA) or “stone mastic asphalt” (Europe);
Response:Thank you for your comment. The authors have chosen the USA expression as “stone matrix asphalt”.
- Figure 3 (a)/Table 2: why didn't you include a tack-coat between PUM-13 and PUM-20 (Structure-I)?
Response:Thank you for your comment. The authors have given the reason of the choose the three typical pavement structures.
The previous research results show that the adhesive strength between PU mixtures and PU mixtures is high, so no adhesive layer is designed between the two layers of PU mixtures [25]. (See Line 146-148)
- Line 164: can you include some references about the “accelerated loading system”?
Response:Thank you for your comment. The authors have added reference [29], which studies on the long-term performance of warm mix asphalt by accelerated loading test. (See Line 510-512)
- Section “4. Results and Discussion”: you should include some further discussion in light of results obtained by other researchers;
Response:Thank you for your comment. The authors have revised the Section of “Results and Discussion”.
(See Line 291-292, Line 323, Line 383-387, Line 390-392, Line 396-399, Line 425-426, Line 429-430.)
- Figure 8 (b): please give note that these values were obtained after 2.0E+4 cycles;
Response:Thank you for your comment. The authors have given the note. The rutting depth of the four mixtures after 10 thousand cycles and 20 thousand cycles rutting is shown in Figure 8 (b). (See Line 291-292)
- Line 272: “Figure 11” or “Figure 8”?
Response:Thank you for your comment. It is written error, the authors have revised. (See Line 285)
- Figure 9: the unit of “Rutting depth” is “cm” instead of “mm”;
Response:Thank you for your comment. It is written mistake, the authors have revised. (See Line 307, Figure 9)
- Section “5. Conclusions”: you can still include or develop further conclusions. At the end of this section (or in section 4), you can also add final comments, like some information on costs and further works;
Response:Thank you for your comment. The authors have revised Section 4. (See Line 291-292, Line 323, Line 383-387, Line 390-392, Line 396-399, Line 425-426, Line 429-430.) The authors have revised Section 5. (See Line 432-434, Line 438-448.)
- A final comment: some of the information/images presented in this manuscript have already been included in another (“Interlaminar Shear Characteristics, Energy Consumption, and Carbon Emissions of Polyurethane Mixtures”, by “Yufeng Bi, Min Sun, Shuo Jing, Derui Hou, Wei Zhuang, Sai Chen, Xuwang Jiao and Quanman Zhao” in “https://doi.org/10.3390/coatings12030400”), so I suggest that you reformulate, as far as possible, some content.
Response:Thank you for your comment. The authors have revised the information/images presented in this manuscript. The paper of “doi.org/10.3390/coatings12030400” focused on Interlaminar Shear Characteristics, Energy Consumption, and Carbon Emissions of Polyurethane Mixtures, the tested structures are different with this manuscript, and the pavement structure of “doi.org/10.3390/coatings12030400” is the structure used in real engineering which are different from the typical structures in this manuscript.